# The Effectiveness of Three-Dimensional Osteosynthesis Plates versus Conventional Plates for the Treatment of Skeletal Fractures: A Systematic Review and Meta-Analysis

**DOI:** 10.3390/jcm12144661

**Published:** 2023-07-13

**Authors:** Iva Ilse Raghoebar, Leander Dubois, Jan de Lange, Tim Schepers, Peter Don Griot, Harald Essig, Frederik Rozema

**Affiliations:** 1Academic Center for Dentistry (ACTA), University of Amsterdam, 1012 WX Amsterdam, The Netherlands; 2Department of Oral and Maxillofacial Surgery, Amsterdam UMC, University of Amsterdam, 1105 AZ Amsterdam, The Netherlands; 3Department of Surgery, Amsterdam Movement Sciences, Amsterdam UMC, University of Amsterdam, 1012 WX Amsterdam, The Netherlands; 4Department of Plastic, Reconstructive and Hand Surgery, Amsterdam UMC, University of Amsterdam, 1012 WX Amsterdam, The Netherlands; 5Department of Oral and Maxillofacial Surgery, University Hospital Zuerich, Frauenklinikstrasse 24, 8091 Zürich, Switzerland

**Keywords:** fractures, conventional osteosynthesis plates, patient-specific implants, preformed osteosynthesis plates, fixation, reduction

## Abstract

Purpose: To assess the difference between preformed anatomically shaped osteosynthesis plates and patient-specific implants versus conventional flat plates for the treatment of skeletal fractures in terms of anatomical reduction, operation time, approach, patient outcomes, and complications. Material and Methods: MEDLINE (1950 to February 2023), EMBASE (1966 to February 2023), and the Cochrane Central Register of Controlled Trials (inception to February 2023) databases were searched. Eligible studies were randomised clinical trials, prospective controlled clinical trials, and prospective and retrospective cohort studies (*n* ≥ 10). Inclusion criteria were studies reporting the outcomes of preformed anatomically shaped osteosynthesis plates and patient-specific implants versus conventional flat plates after treating skeletal fractures. Outcome measures included anatomical reduction, stability, operation time, hospitalisation days, patients’ outcomes, and complications. Two independent reviewers assessed the abstracts and analysed the complete texts and methodologies of the included studies. Results: In total, 21 out of the 5181 primarily selected articles matched the inclusion criteria. A meta-analysis revealed a significant difference in operation time in favour of the preformed anatomical plates and patient-specific implants versus conventional plates. Significant differences in operation time were found for the orbital (95% CI: −50.70–7.49, *p* = 0.008), upper limb (95% CI: −17.91–6.13, *p* < 0.0001), and lower limb extremity groups (95% CI: −20.40–15.11, *p* < 0.00001). The mean difference in the rate of anatomical reduction in the lower limb extremity group (95% CI: 1.04–7.62, *p* = 0.04) was also in favour of using preformed anatomical plates and patient-specific implants versus conventional plates. Conclusions: This systematic review showed a significant mean difference in surgery time favouring the use of preformed anatomical plates and patient-specific implants for orbital, upper, and lower limb extremity fractures. Additionally, preformed anatomical plates and patient-specific implants in the lower limb group result in a significantly higher rate of anatomical reduction versus conventional flat plates.

## 1. Introduction

Adequate treatment of skeletal fractures is one of the pillars of trauma care [1]. Fractures result from forces that cannot be withstood by the strength of the bone tissue [2]. Sport-related injuries, traffic accidents, assaults, and falls have been reported as common causes of fractures [2,3,4]. The incidence of fractures is increasing rapidly worldwide due to progressive demographic transformations and rising life expectancy [5]. A fracture results in a loss of bone integrity and a break in structural continuity at the fracture site [3,6,7]. The sequence of systemic and specific tissue responses during fracture healing involves inflammation, granulation, callus formation, and callus remodelling [8].

Five feasible treatment options are generally recognised for fractures: conservative, immobilization with a cast (splints or braces), skeletal traction, open reduction with internal fixation, and open reduction with external fixation [3]. The primary goal of fracture management is to re-establish the structural integrity of the fractured bone, to restore normal function [9,10,11]. Additionally, the four recognised principles of osteosynthesis are fracture reduction, fracture fixation, preserving the blood supply, and early mobilization [12]. Unfortunately, a uniform consensus regarding fracture treatment standards has not been reached. One of the main reasons is the large variability in the type of fractures and the type of bone involved. Which treatment option is chosen depends on the amount (and direction) of muscle traction and load influencing the fracture, the pattern of the fracture, the need for joint immobilization, and the interfragmentary support [13,14,15,16].

The conventional treatment method for bone fractures was to manage them conservatively for a long time. However, the widespread availability of antibiotics has played an essential role in the shift to surgical treatment [17]. Additionally, the development of osteosynthesis material in the previous century has revolutionised fracture management. Steel, gold, and nickel-coated steel were commonly used at the beginning of the 20th century [18]. Later, titanium plates were introduced to overcome some of the limitations of conventional materials, the advantages being less corrosion, metallosis, broken plates, loose screws, and more stability [18]. Newer developments are focusing on biodegradable plates since they can achieve adequate stability in defined cases and have the ability to resorb gradually [19,20,21]. The above-mentioned improvements in operative management have allowed surgical fracture treatment to emerge as a viable option worldwide.

Adequate anatomical reduction and a stable position until the fracture has healed are essential in the surgical treatment of fractures [11,22]. If open reduction with internal fixation (ORIF) is indicated, the standard treatment mainly involves internal fixation with titanium or stainless steel plates and screws to achieve stability. The straight commercial titanium plate is contoured intraoperatively to the individual bone’s geometry, size, shape, and complex fracture pattern [23,24,25]. Extensive bending may be mandatory to individualize the titanium plates [25]. However, the residual stress of the plate increases the risk of fatigue failure of the material [23,25,26].

To overcome extensive preparation, anatomically shaped 3D plates were developed. The shape of these plates is based on the statistical shape model (SSM). The SSM represents the average three-dimensional shape and modes of variation in a population [27,28]. Even though these off-the-shelf plates offer perspective, they do not fit the entire population and often require minor adjustments during surgery [28]. Nevertheless, some surgeons prefer using conventional plates due to their easy accessibility and relatively low costs [2,26].

The availability of computed tomography (CT) and the advanced technical option of computer-aided surgery (CAS) have opened new ways to prevent or reduce the number of surgical approaches and complications and avoid further plate adjustments [25,29]. Introducing patient-specific implants (PSIs) based on a patient’s CT has revolutionised trauma care. New design options can be implemented to optimise these plates; this is a major advantage compared with conventional plates [29]. On the downside, logistical obstacles might arise since these plates are not off-the-shelf, and there is a potential economic limitation as they are far more expensive than traditional plates.

The importance of providing sufficient evidence is underscored by the fact that using preformed anatomically shaped osteosynthesis plates and PSIs can potentially improve the reduction accuracy, achieve greater stability, and minimise surgical approaches and complications [25,30]. Non-optimal repositioning of fractures can lead to severe complications, which can have a significant impact on health and quality of life [31,32]. Therefore, the present systematic review aimed to evaluate whether preformed anatomically shaped osteosynthesis plates and PSIs are preferable over conventional flat plates to treat skeletal fractures in terms of anatomical reduction, stability, operation time, hospitalisation days, patient outcomes, and complications.

## 2. Methods

### 2.1. Protocol Development

A protocol was developed a priori to answer the following question: Are there outcome differences between preformed anatomically shaped osteosynthesis plates (both stock implants and PSIs) and conventional flat plates in the management of fractures, especially regarding anatomical reduction, stability, operation time, hospitalisation days, patient outcomes, and complications? This systematic review fulfilled the Preferred Reporting Items for Systematic Reviews and Meta-Analyses (PRISMA) guidelines [33]. 

### 2.2. Information Sources and Search Strategy

The search strategy was developed with the help of a biomedical information specialist according to the syntax rules of each database (Table 1). An extensive literature search of the following electronic databases was conducted on 1 February 2023: MEDLINE (1964–2023), EMBASE (1947–2023), and the Cochrane Central Register of Controlled Trials (CENTRAL; inception to 2023). The automated search was completed by hand-searching the references of eligible review articles and relevant studies for additional valuable publications.

### 2.3. Eligibility Criteria

To be eligible, the studies had to meet the following criteria: Type of Patients or population: adult patients with skeletal fractures requiring surgical fixation treatment.Type of intervention: fixation with preformed osteosynthesis plates (including PSIs).Comparison of the control group: fixation with conventional plates.Primary outcome: anatomical reduction.Secondary outcomes: stability and complications, days spent in the hospital, operation time, and patient satisfaction.Study design: randomised clinical trials (RCTs), prospective controlled clinical trials (CCTs), and prospective and retrospective cohort studies.Study language: there were no language restrictions.

Exclusion criteria:Studies with fewer than 10 patients;Use of biodegradable systems (only materials such as titanium or RVS were included);Pathological fractures;Case reports, case series, experts’ opinions, conference abstracts, letters to the editor, animal studies, reviews, and systematic reviews.

### 2.4. Screening Methods

Two independent reviewers (IR and LD) screened the titles and abstracts for inclusion eligibility. Full-text documents were obtained of all the articles meeting the inclusion criteria. An article underwent full-text assessment in case of doubt or insufficient information, which was performed independently by the two reviewers, and any disagreement between the two was resolved by a discussion. In case of a persistent disagreement regarding inclusion, a third reviewer (FRR) could be consulted.

The interrater reliability of the title, abstract screening, and full-text analysis were quantitatively measured with Cohen’s kappa coefficient (κ) and percentage of agreement. The non-English articles were translated by a native speaker of both the language of the concerned article and English. If the full-text article could not be obtained, our university’s information analyst and the study’s researchers were contacted.

### 2.5. Data Extraction

A standardised pre-specified form extracted the following data from the included studies: author(s), year of publication, study design, country, number of patients, males/females, mean age, age range, follow-up, radiographic assessment/imaging technique, clinical assessment, fracture site, type of plate(s), material of the plates, stability, bone union, nonunion, delayed union, malunion, rate of anatomic reduction, screw loosening, hardware failure or plate palpability, operation time, union time, infection, revision surgery, patient-reported outcome measures (patient satisfaction), and specific fractured site parameters.

In this review, the rate of anatomic reduction refers to a percentage of fractures that have been accurately repositioned to their anatomically correct alignment. Stability is defined as ‘a reduced ZMC fracture, which remains in a stable anatomical position over time’.

### 2.6. Evaluation of Study Quality and Risk of Bias

The methodological quality of the included studies was assessed independently by the two reviewers (IR and LD). The risk of bias in the randomised studies was evaluated using the seven domains of the Cochrane collaboration tool for RCTs (RoB 2.0) [34]. The Methodological Index for Non-Randomized Studies (MINORS) was used to assess the methodological quality of the non-randomised studies [35]. Disagreement between the two reviewers was discussed in a meeting, and a third reviewer (FRR) could be consulted if required. Cohen’s kappa coefficient (κ) and percentage of agreement were measured.

### 2.7. Statistical Analysis 

The statistical software package ‘Meta-analysis’ (Review Manager 5.4) was used. The events and totals of the dichotomous outcomes were used to calculate the odds ratio (OR) and corresponding 95% confidence intervals (CIs). A random-effects model was applied to the continuous outcomes to calculate the mean differences and corresponding 95% confidence intervals (CIs). To assess the statistical heterogeneity among studies, the I^2^ was calculated, with no heterogeneity being quantified by 0%, mild heterogeneity by <30%, moderate heterogeneity by 30–60%, and notable heterogeneity by >60%. To calculate the overall statistical heterogeneity between the included studies, a meta-regression analysis (random-effects model) was performed.

## 3. Results

### 3.1. Study Selection

The primary search on 1 February 2023 resulted in 2480 hits for Medline, 2293 hits for Embase, and 408 hits for Cochrane (Figure 1). Four additional records were identified by manually checking the reference list of the eligible reviews and studies. A total of 3416 titles and abstracts were screened after eliminating duplicate records, of which 3306 articles were excluded because they did not meet the inclusion and exclusion criteria. Cohen’s kappa coefficient (κ) and percentage of agreement for the titles and abstract screening were 0.61 and 94.5, respectively.

This approach resulted in 110 manuscripts for full-text analysis. Of these, 21 articles fulfilled the inclusion criteria and were used for quantitative syntheses. Cohen’s kappa coefficient (κ) and percentage of agreement were 0.72 and 87.2%, respectively. Any disagreement was resolved by a discussion and, after a consensus meeting, 24 articles about mandibular fractures were excluded. One of the screened full-text manuscripts was written in French; this article was translated into English by a fluent French and English speaker. The authors of five non-available full-text articles were contacted. These five articles were excluded because the authors did not respond.

### 3.2. Study Characteristics

The characteristics of the studies used for the qualitative syntheses are depicted in Table 2. The search yielded seven RCTs, four prospective cohort studies, and ten retrospective cohort studies. Seven studies evaluated the outcome difference between the intraoperative bending of titanium mesh and pre-contoured implants in the surgical repair of orbital fractures [36,37,38,39,40,41,42]. Another seven studies compared the treatment outcome of conventional flat plates versus anatomical pre-contoured plates in patients with upper limb fractures [43,44,45,46,47,48,49]. Furthermore, seven studies assessed the difference between conventional plates and preformed plates in the surgical treatment of lower limb fractures [50,51,52,53,54,55,56]. Of the seven included RCTs, four studies reported lower limb fractures and three studies upper limb fractures [45,46,49,52,53,54,55].

The number of patients varied from 10 [37] to 93 [42]. The total of included patients was 928, of which 464 were treated with conventional plates and 464 with preformed anatomical plates. Nineteen out of the twenty-one included studies reported follow-up times (Table 2) varying from 6 weeks [37] to 24 months [43].

All the orbital group studies used pre- and postoperative 3D imaging (CT-scan) [36,37,38,39,40,41,42]. Four upper limb fracture studies used 3D imaging preoperatively [45,46,48,49], whereas the other three used 2D imaging preoperatively [43,44,47]. Shuang et al. (2016) obtained a CT scan postoperatively, Chen et al. (2019) made a CT scan based on the surgeon’s preference, and You et al. (2016) only took a CT scan of the experimental group [45,46,48]. Kong et al. (2020), Ellwein et al. (2019), DelSole et al. (2019), and Tang et al. (2012) obtained 2D imaging rather than 3D imaging postoperatively [43,44,47,49]. In the lower limb group, Zhang et al. (2019), Zheng et al. (2018), and Zheng et al. (2017) used 3D imaging preoperatively (CT scan) and 2D imaging postoperatively (anteroposterior, lateral, or oblique X-ray) [54,56,57]. Zyskowski et al. (2021), Park et al. (2020), and Bilgetekin et al. (2019) reported data from 2D images made preoperatively and postoperatively [50,51,52]. Tsukada et al. (2013) did not specify if any form of imaging was used preoperatively. However, they used 2D imaging intraoperatively and postoperatively [55].

Six orbital fracture studies reported that the conventional and experimental groups were treated with titanium material. The seventh orbital fracture study reported that titanium was used in the conventional group; the experimental group received ultra-high molecular weight polyethylene (UHMW-PE), dioxide zirconium (ZrO), and rapid prototyping (RP) titanium [42]. Among the lower limb fracture group, one study used titanium in both the conventional and experimental groups [55], and one study used steel in the experimental group [54]. Regarding the upper limb fracture group, one study used steel in both the conventional and experimental groups [45]. Eleven studies did not report the type of material used [43,44,46,47,48,49,50,51,52,53,56].

### 3.3. Assessment of Methodological Quality

The results of the methodological quality assessment of the included RCTs are summarised in Figure 2. Cohen’s kappa coefficient (κ) for the RoB2 and MINORS domains ranged between 0.82 and 1.0 (percentage of agreement: 87–100%) and between 0.89 and 1.0 (percentage of agreement: 90–100%), respectively. The risk of bias ranged from medium to low. Three studies had a medium score due to some concerns with the ‘Bias arising from the randomization process’ domain [45,46,52]. Zyskowski et al. (2021), Shuang et al. (2016), and You et al. (2016) did not report if a random component was used in the sequence generation process, which resulted in some concerns. In the remaining four RCTS, the risk of bias had a low score [49,53,54,55].

The quality of the non-randomised studies is depicted in Figure 3. All the retrospective and prospective studies failed to perform an unbiased assessment of the study or perform a sample size calculation a priori. Five retrospective studies had lost more than 5% of the patients to follow-up [39,43,44,47,56]. One prospective study and two retrospective studies did not report the inclusion criteria [36,40,42]. Both Fan et al. (2017) and Momjian et al. (2011) gave some baseline characteristics but did not make any statements regarding the baseline equivalences of the groups [36,39]. The follow-up time was not mentioned by Sigron et al. (2020) and Fan et al. (2017). Overall, all the non-randomised studies had a clearly stated aim, appropriate endpoints, an adequate control group, and contemporary groups, and their statistical analyses were adequate.

### 3.4. Outcome Measures

#### 3.4.1. Orbital Fractures

The seven included orbital fracture articles were prospective and retrospective cohort studies. Data were extracted for qualitative analysis, and two studies’ data could be meaningfully pooled for a meta-analysis (Section 3.5 meta-analysis).

Scolozzi (2011) and Scolozzi et al. (2010) reported three-dimensional anatomical placement after orbital volume restoration with non-preformed and three-dimensional preformed plates [37,38]. All control and experimental group patients showed anatomical three-dimensional placement postoperatively (Table 3) [37,38]. Operation time was reported in three articles (Table 4) [36,41,42]. Zielinkski et al. (2017) noted longer operation times for the conventional group than the experimental group. However, no significant difference was observed between the two groups [42]. Both Sigron et al. (2020) and Fan et al. (2017) showed a significant mean difference in surgery time favouring the use of preformed anatomical plates [36,41]. The patients receiving preformed anatomical plates needed significantly lower placement times than those with conventional plates [40]. Sigron et al. (2020) and Zielinkski et al. (2017) reported no significant differences in hospitalisation days [41,42].

The compared postoperative mean orbital volumes and mean absolute volume differences between both treatment groups did not differ significantly, according to Momjian et al. (2011) and Scolozzi (2021) (Table 5) [37,39]. At the same time, Sigron et al. (2020) found a significant difference between the conventional and experiential groups. Sigron et al. (2020) also reported a statistically significant difference (*p* = 0.002) in the mean absolute volume of the preoperative non-fractured orbital and the postoperative reconstructed orbital in the conventional group [41]. In contrast, the difference in the mean absolute volume in the experimental group with anatomically preformed plates was not significant [41].

Wilmowsky et al. (2020) reported no significant differences between the conventional and experimental groups with regard to the congruence of the infraorbital rim (Table 5) [40]. None of Scolozzi et al. (2011) and Momjian et al.’s (2011) patients presented enophthalmos or limitations of the inferior oblique muscles [37,39]. Significantly lower scores were observed by Fan et al. (2017) in the anatomically preformed plate group’s maximum width, depth, and area between the fracture zone and implant compared to the conventional flat plate group [36]. Additionally, comparing the two groups, Fan et al. (2017) indicated that the experimental group had significantly better enophthalmos and superior sulcus deformity outcomes [36].

#### 3.4.2. Upper Limb Fractures

Seven articles reported on upper limb fractures, of which three publications were RCTs and four were retrospective cohort studies. Data were extracted from the seven articles for qualitative assessments. Four studies could be meaningfully pooled for a meta-analysis (Section 3.5 meta-analysis).

Each publication’s outcome measure of variable stability is depicted in Table 6. Ellwein et al. (2019), DelSole et al. (2016), and You et al. (2016) reported that all the patients in the conventional and experimental groups achieved bone union [43,45,47]. Tang et al. (2012) noted that none of the patients had a loss of reduction [44]. Kong et al. (2020) reported a 6.25% loss of reduction in both the conventional and experimental groups [48,49]. Time to union was observed after 8.50 ± 1.22 and 11.76 weeks in the conventional group and after 8.36 ± 1.00 and 12.93 weeks in the experimental group [44,45]. Even though there was variability in loss of reduction and time to bone union, no statistical difference was observed [44,45,48,49].

**Table 6 jcm-12-04661-t006:** Outcome measure: stability; fractured site: upper limb. Abbreviation: N: number of patients, SD: standard deviation.^1^ Loss of reduction was assessed with radial inclination, radial height and volar tilt by Chen et al (2019), reported in Table 7. ^2^ Stability was assessed with range of motion (ROM) by Chen et al (2019), reported in Table 7.

Parameter I: Stability and Reduction	Bone Union (N/Total Population)	Loss of Reduction (N/Total Population)	Time to Bone Union (Mean Weeks ± SD, Total Population)	Stability (N/Total Population)
Upper Limb Fractures				
Kong (2020) [49]		2D: 1/16; 6.25% 3D: 1/16; 6.25%		
Ellwein (2019) [43]	2D: 18/18; 100% 3D: 10/10; 100%			
Chen (2019) [48]		^1^		^2^
DelSole (2016) [47]	2D: 14/14; 100% 3D: 8/8; 100%			
You (2016) [45]	2D: 32/32; 100% 3D: 34/34; 100%		2D: 8.50 ± 1.22 (32) 3D: 8.36 ± 1.00 (34)	
Tang (2012) [44]		2D: 0/17; 0% 3D: 0/16; 0%	2D: 11.76 (range: 9–19) (17) 3D: 12.93 (range: 10–17) (16)	

Loosening of screws was reported for 4% of the patients in the conventional group and 0% of the patients in the experimental group; this difference was not significant (Table 8) [43]. None of the patients in either treatment group presented necrosis or nerve injury [48,49]. No significant difference in infection rate was revealed between the conventional and experimental groups [46,47,48,49].

There was no significant difference in revision surgery rate between both treatment groups (Table 9) [43,47,48]. No hardware failure was reported for either the conventional or experimental group [43,44,47]. Ellwein et al. (2019) and Tang et al. (2012) observed that the conventional group’s operation time was not significantly shorter [43,44]. These findings were not in agreement with the statistically shorter operation time observed in the experimental group by Kong et al. (2020), You et al. (2016), and Shuang et al. (2016) [45,46,49]. Both operation time and intraoperative blood loss and intraoperative fluoroscopy times were significantly shorter and lower in the experimental group than the conventional group [45,46,49]. The patients’ satisfaction was 96% in the conventional group and 91% in the experimental group [43]. This difference was not significant.

Chen et al. (2019) showed that the preformed anatomical plates group had a significantly better modified mayo wrist score (MMWS), range of motion: extension/flexion and pronation/supination, and Disabilities of the Arm, Shoulder, and Hand (DASH) score (Table 7) [48]. Even though Ellwein et al. (2019) and Shuang et al.’s (2016) results were not significant, the rest of the data were generally in line with Chen et al.’s (2019) results, except for the variable range of motion, such as pronation/supination in the Shuang et al. (2016) study [43,46,48]. Even though Kong et al. (2020) reported lower DASH scores for the experimental group after six months [49] and Tang et al. (2012) reported lower DASH scores for the conventional group after one year [44], their scores were not significantly different [44,49].

#### 3.4.3. Lower Limb Fractures

A total of seven lower limb fracture publications were selected, including four RCTs and three retrospective cohort studies. After a quality assessment, only four studies’ data could be meaningfully pooled for a meta-analysis (Section 3.5 meta-analysis).

All of the Zyskowski et al. (2021), Park et al. (2020), Bilgetekin et al. (2019), Zhang et al. (2019), and Zheng et al. (2018), and Zheng et al. (2017) patients recovered with no signs of nonunion (Table 10) [50,51,52,53,54,56]. No significant differences in delayed union and malunion were observed between the conventional and experimental groups [50,53,56]. After 12 months, 19 of the 21 patients achieved bone union in the conventional group and 23 of the 24 patients in the experimental group; this difference was not significant [55]. Zheng et al. (2018) showed a significantly higher rate of anatomical reduction in the experimental group than the conventional group [53]. In line with this result, Zhang et al. (2019) also reported a higher rate of anatomical reduction in the experimental group, although no significant difference was seen [56].

The plate was palpable in two patients (10%) from the experimental group and in none of the patients (0%) from the conventional group (Table 11) [52]. Zyskowski et al. (2021) reported a 4% rate of swelling and deep vein thrombosis in the conventional group, while no swelling and deep vein thrombosis were observed in the experimental group [52]. However, no significant difference was observed between both groups regarding plate palpability, screw loosening, swelling, infection rate, and deep vein thrombosis [50,51,52,53,54,55,56].

The fracture to union time was shorter in the 3D group than the 2D group, although there was no significant difference between both groups (Table 12) [53,54,56]. A significantly shorter operation time was noted for the experimental group than the conventional group [50,53,54,56]. The number of fluoroscopies and intraoperative blood loss in the anatomically preformed plate group was significantly less than the conventional group [53,54,56].

The experimental group presented with a greater range of plantar and dorsiflexion motion than the conventional group; nevertheless, no clinical statistical difference was observed between both groups (Table 13) [52,53,56]. The American Orthopaedic Foot and Ankle Society (AOFAS) score was reported as higher in the experimental group by Zhang et al. (2019), Zheng et al. (2018), and Zheng et al. (2017) [53,54,56]. Contrarily, Bilgetekin et al. (2019) reported a higher AOFAS score in the conventional group [51]. However, the AOFAS score difference was insignificant between both treatment groups [51,53,54,56].

### 3.5. Meta-Analysis

#### 3.5.1. Primary Outcome: Adequate Anatomical Reduction

Both Scolozzi (2011) and Scolozzi et al. (2010) reported three-dimensional anatomical placement in all the patients from both groups [37,38]. Although Tang et al. (2012) and Kong et al. (2020) assessed reduction in the upper extremity group, Tang et al. (2012) reported a total zero event [37,38]. Therefore, pooling of the data was impossible for the orbital and upper limb extremity groups. A meta-analysis could only be performed with data from the lower limb extremity group.

A meta-regression analysis showed a statistically significant difference in favour of using preformed anatomical plates versus conventional plates. The forest plot of the odds ratio meta-analysis is depicted in Figure 4. The anatomically preformed plate group’s reduction rate was 2.86 higher (Figure 4: 95% CI: 1.04–7.62, *p* = 0.04, I^2^ = 0%, *n* = 2 studies) than that the conventional plate group.

#### 3.5.2. Secondary Outcome: Operation Time

Data were extracted from studies reporting on operation time in the orbital, upper, and lower extremity groups and could be meaningfully pooled for a meta-analysis.

Figure 5 shows the forest plots of the random and fixed effects meta-analysis. The mean difference in the orbital group’s operation time was 29.09 min (Figure 5a: 95% CI: –50.70–7.49, *p* = 0.008, I^2^ = 70%, *n* = 2 studies), which was in favour of the use of preformed plates versus conventional plates. A significant mean difference of 12.02 min (Figure 5b: 95% CI: −17.91–6.13, *p* < 0.0001, I^2^ = 59%, *n* = 4 studies) was revealed for the upper limb group, which was also in favour of the use of preformed plates versus conventional plates. The mean difference in the lower limb extremity group was 17.76 min (Figure 5c: 95% CI: −20.40–15.11, *p* < 0.00001, I^2^ = 0%, *n* = 4 studies), which was also in favour of the use of preformed plates versus conventional plates.

Overall, the results concerning operation time in this systematic review favoured the preformed versus conventional plates. Thus, a significantly shorter operation time was reported in the experimental group with preformed anatomical plates in the orbit, upper, and lower limb extremities.

## 4. Discussion

This systematic review aimed to assess and evaluate the difference in the effect of preformed anatomically shaped osteosynthesis plates and PSIs versus conventional flat plates on treating skeletal fractures in terms of anatomical reduction, operation time, approach, patient outcome, and complications.

The industry has reported that preformed anatomically shaped osteosynthesis plates are a powerful and superior treatment modality for patients with fractures. According to the industry, preformed anatomically shaped plates can realise more stability and reduce surgery time, leading to simplified surgical operations and better clinical outcomes [30]. Surgeons have widely accepted and recognised the benefits of preformed anatomical plates. However, these statements are based on assumptions and limited clinical research, so more clinical evidence about preformed anatomical plates is still required.

Our findings in this systematic review are partly consistent with recent literature, which claims that anatomically preformed plates have potentially beneficial clinical outcome effects. We show that the conventional flat plate group needs significantly more surgical time than the anatomically preformed plates group. The meta-analysis revealed a significantly favourable result in favour of the use of anatomically preformed plates in the orbit, upper extremity, and lower extremity fracture groups. This systematic review also points to a significantly higher rate of anatomical reduction in the experimental group than the conventional group. However, the meta-analysis only showed a significant difference in the lower limb group.

Most of the reviewed studies reported a longer operation time in the conventional group compared to the experimental group, the exception being Ellwein et al. (2019) and Tang et al. (2012) [43,44]. Both of these studies did not describe a significant difference and could not give a reasonable explanation for the contradictory results [43,44]. A possible explanation could be unfamiliarity with the surgical procedure for the preformed anatomical plate. Tang et al. (2012) was not included in the meta-analysis because no standard deviation was reported. Although Ellwein et al.’s (2019) contrary results were included in the analysis, they did not influence the outcome.

Adequate anatomical reduction is a critical factor for the successful surgical treatment of fractures and was the primary outcome of this systematic review. In retrospect, most of the included studies did not document or report this variable. Hence, only the data from the lower limb group could be meaningfully pooled. However, all the studies reported on sub-variables such as nonunion, delayed union, malunion, anatomical 3D placement, and loss of reduction. One could argue that the orbit variables and absolute volume difference in the orbit group also represent reduction accuracy. No concluding remarks regarding the sub-variables can be made in this systematic review. A comment needs to be made about the adequate anatomical reduction outcomes and the sub-variables determined with radiographic images; the studies’ radiographic algorithms seem to vary significantly in terms of the type of image, timing, and follow-up time. Such differences in imaging can potentially influence the outcomes.

It needs to be noted that there were a few limitations to our study. The level of experience possessed by the surgeons was not taken into consideration in this review. It needs to be noted that the experience of the surgeon can influence both the rate of anatomical reduction and the duration of the surgical procedure. Unfortunately, anatomical reduction and operation time were the only variables that could be meaningfully pooled in this review. The main reason for this is that the number and prevalence of the outcome events in both treatment modality groups were either too small or too big, which inevitably makes the statistical power of the meta-analysis low. Additionally, no indications could be identified in this systematic review to suggest that the type of plate played a role in stability and complications, hospitalisation days, patient satisfaction, size, and the number of approaches. Theoretically, a reduction in operation time can positively affect blood loss, postoperative complications, and infections [49,54]. However, this relation cannot be confirmed in this systematic review.

The evolution of three-dimensional scanning and the implementation of these scans in plate designs is a fundamental principle of this systematic review [46]. It was stated that three-dimensional planning significantly improves the accuracy and predictability of implant positioning [58]. Our study described and tested the difference in results between plates. However, we did not test the influence of using a three-dimensional scan and virtual planning. Ten studies used a preoperative three-dimensional scan for diagnostic purposes or to generate a three-dimensional model [36,39,40,42,45,46,49,53,54,56]. Such a three-dimensional model is primarily used to fabricate a patient-specific implant [46,53,54] as well as show the precise anatomical structure and facilitate visualization of the structural feature of the fracture [51,52]. This model allows for the adaptation of a detailed preoperative plan and the possibility to virtually check the fit, shape, and size of the preformed implant [58,59]. The virtual model can be printed and then used during surgery as a reference for adequate anatomical reduction [51,52]. These three-dimensional printed models create more understanding of and compliance with patients [52]. We did not consider the above-mentioned beneficial factors of three-dimensional scanning and modelling.

In our study, no difference was made between preformed anatomically shaped osteosynthesis plates and patient-specific implants (PSIs), which means each type of preformed plate with different preparation and design options could not be analysed separately. In addition, preformed anatomical plates are based on the statistical shape model (SSM), and PSIs are based on patients’ CTs [27,28]. Preformed anatomical osteosynthesis plates might still require to be manually contoured intraoperatively due to variabilities in anatomical features. PSIs, on the other hand, are designed to fit optimally and do not need intraoperative changes. The data were analysed as one group, which could have resulted in bias. This may have contributed to the outcomes because the plate type can influence the operation time, course of recovery, and clinical outcomes.

It must be noted that the treatment algorithms of the studies in this review varied significantly regarding plate material and radiographic imaging. Eleven of the twenty-one included studies did not report the type of material used [43,44,46,47,48,49,50,51,52,53,56]. We do not expect that these studies use biodegradable osteosynthesis material, but we cannot rule the possibility out. This could have potentially influenced our review’s results since the material properties of biodegradable plates differ from more conventional materials, such as titanium and RVS [60,61]. Biodegradable material can cause late tissue responses, which may have a negative impact on the mechanical characteristics of the plate, causing decreased stability [20,62]. Another limitation of this review is the variability in radiographic algorithms between the studies; the pre- and postoperative imaging differed within and between studies. One noticeable observation is the fact that six studies used 3D imaging preoperatively, whereas they applied 2D imaging postoperatively [45,48,49,54,56,57]. Three-dimensional imaging is favourable for indication and evaluation purposes [63,64]. Comparisons between the three-dimensional preoperative scans and planning can be made more accurately if the preoperative and postoperative scans both use 3D information. These differences impede comparisons between the studies.

Our systematic review included articles that used the contralateral healthy unfractured site as a reference for the mirror model. The three-dimensional model is based on the mirror model [51,52]. There were reported structural anatomical differences between contralateral bones, especially in bone quantity and organisation [65]. Asymmetry between contralateral sides can result in a non-optimal fit of the implant, requiring intraoperative adjustments to make anatomical placement possible. On the other hand, it has been reported that left-versus-right anatomical differences in non-traumatised orbital cavities and zygoma are clinically minor [29,66]. In light of these findings, preoperative planning with the mirroring technique can offer a lot of perspective for orbital reconstructions [66]. Based on the statements mentioned above, the anatomical differences seen by us were negligible in the orbital group but could have played a more prominent role in the upper and lower limb groups. However, none of the studies reported major intraoperative adjustments in the experimental group, so we do not expect this to influence the results.

It is essential to state that our study was designed a priori to give insight into the difference between conventional and preformed anatomical plates in the treatment of orbital, mandibula, zygoma, upper extremity, and lower extremity fractures. We did not analyse mandibula or zygoma fractures. Twenty-four articles about mandibular fractures were excluded after reading the full text because they did not meet the inclusion criteria. A critical inclusion criterion was that conventional flat plates were compared to preformed anatomical plates. Even though those articles stated ‘3D anatomical plates’ had been applied in the experimental group, these plates cannot, in our opinion, be classified as anatomically preformed at a three-dimensional level. The term ‘three-dimensional’ is a misnomer in those excluded studies [67,68]. The plate contributes to stability and resists shearing, bending, and torsional forces in three dimensions [67]. No articles were found that compared conventional and preformed plates to manage zygoma fractures. Hence, no statements or conclusions can be made in this systematic review about the difference in outcomes between conventional and preformed anatomical plates applied to mandibula and zygoma fractures.

Following the suggestions for further research to evaluate the outcome of anatomically preformed plates inserted to treat skeletal fractures, our systematic review shows that using anatomically preformed plates can significantly reduce the operation time in the orbital, upper, and lower extremity group and can significantly help improve the surgical reduction in the lower limb fracture group. Three-dimensional imaging and computer-aided navigation are fundamental principles of these preformed plates. Questions arise if the high device costs, limited availability of CBCT scans, and complexity of the operation weigh against the potential benefits of preformed anatomical plates [46]. These observations need further research. Conducting well-designed RCTs with large sample sizes, sufficient follow-up durations, and uniform imaging algorithms is mandatory. Nowadays, studies tend to focus on clinical outcomes, but patient satisfaction is just as important and should be documented in future research. Optimization of the treatment algorithm holds a grand promise for improving surgical-, clinical-, and patient-reported outcomes.

## Figures and Tables

**Figure 1 jcm-12-04661-f001:**
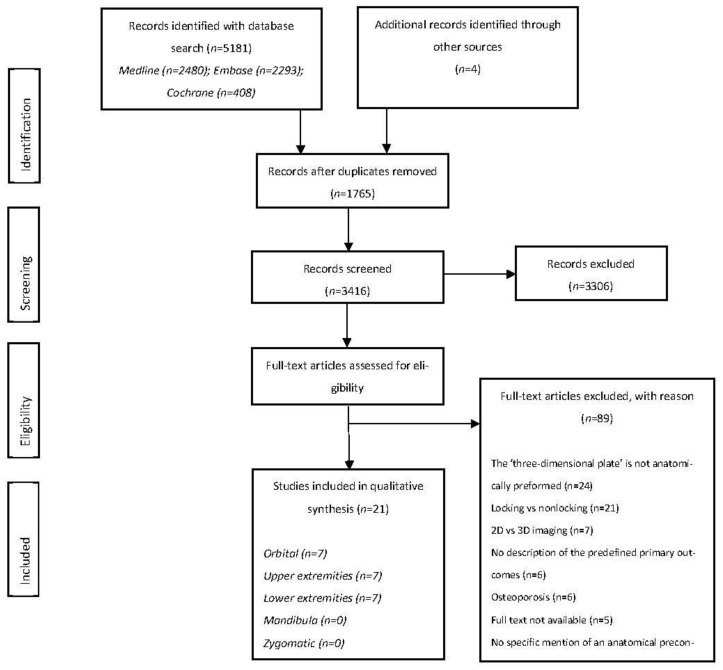
Algorithm of the study selection procedure.

**Figure 2 jcm-12-04661-f002:**
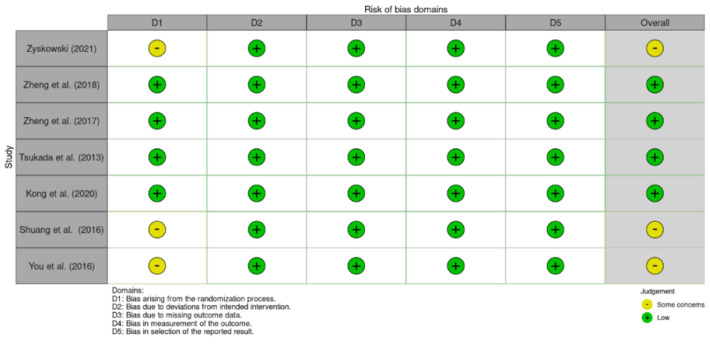
Risk-of-bias assessment of the included RCTs (RoB2) [45,46,49,52,53,54,55].

**Figure 3 jcm-12-04661-f003:**
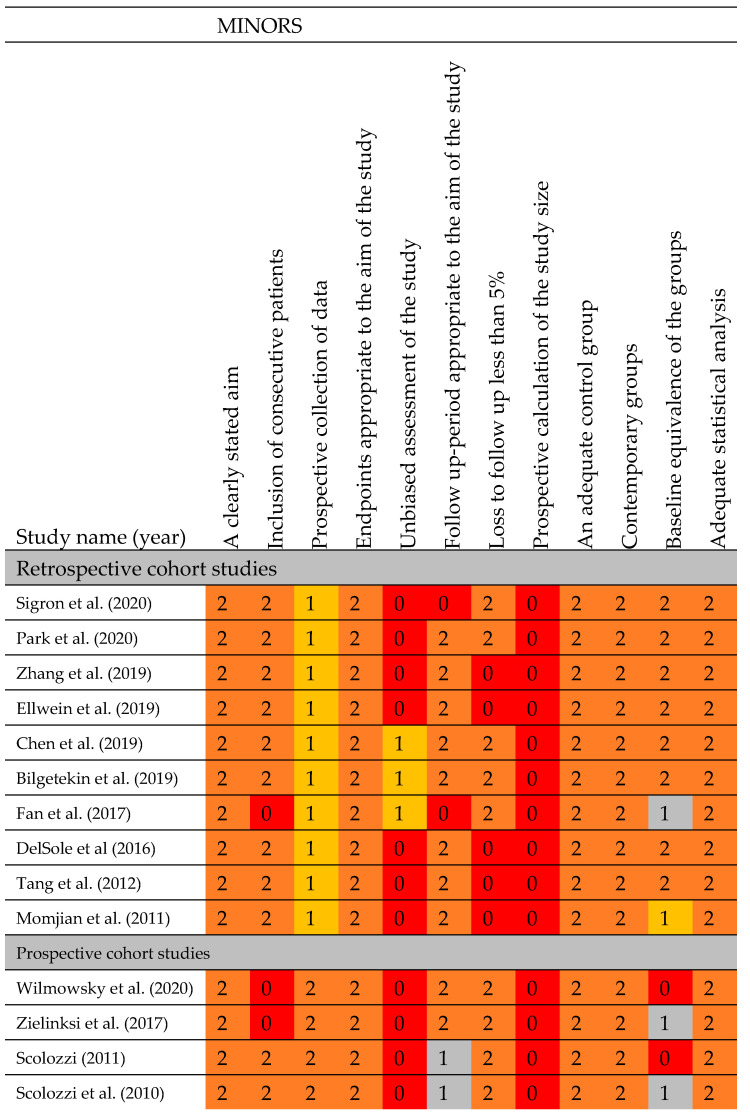
Risk-of-bias assessment of the included prospective and retrospective studies (MINORS) [36,37,38,39,40,41,42,43,44,47,48,50,51,56]. Abbreviations: 0: not reported; 1: reported but inadequate; 2: reported and adequate.

**Figure 4 jcm-12-04661-f004:**
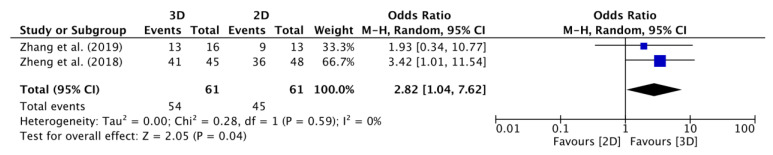
Forest plots of the odds ratio meta-analysis: rate of anatomical reduction in the lower limb group (preformed anatomical plates: 3D versus conventional 2D plates) [53,54]. Abbreviations: 3D: three-dimensional; 2D: two-dimensional; CI: confidence interval.

**Figure 5 jcm-12-04661-f005:**
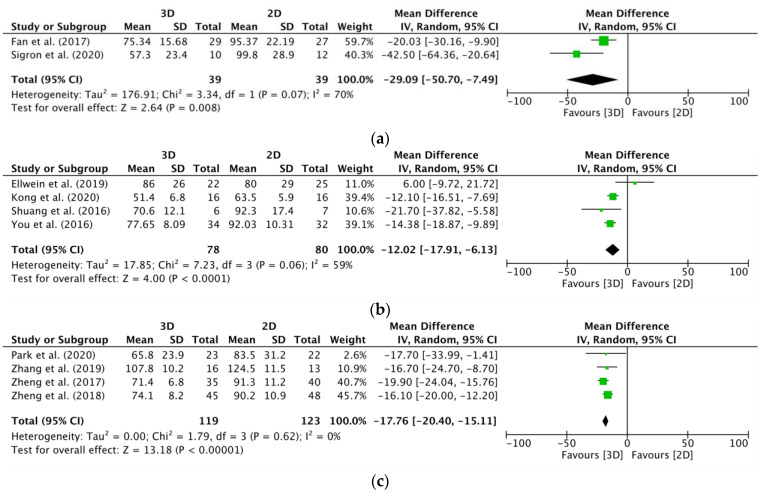
Forest plots of the mean difference (min) in operation time: preformed anatomical plates (3D) versus conventional plates (2D). (**a**) Orbital group: 3D versus 2D [36,41]. (**b**) Upper extremity group: 3D versus 2D [43,46,49,45]. (**c**) Lower extremity group: 3D versus 2D [50,53,54,56]. Abbreviations: 3D: three-dimensional; 2D: two-dimensional; SD: standard deviation; CI: confidence interval.

**Table 1 jcm-12-04661-t001:** Search Strategy.

Database	Strategy
MEDLINE	(“Fractures, Bone”[Mesh] OR “Fracture Fixation”[Mesh] OR fractur*[tiab]) AND (“Fracture Fixation, Internal”[Mesh] OR “Bone Plates”[Mesh] OR fixat*[tiab] OR osteosynth*[tiab] OR plate*[tiab] OR print*[tiab]) AND (“Wrist Injuries”[Mesh] OR “Wrist Joint”[Mesh] OR “Wrist”[Mesh] OR “Radius Fractures”[Mesh] OR “Ulna Fractures”[Mesh] OR “Jaw Fractures”[Mesh] OR “Mandibular Injuries”[Mesh] OR “Orbital Fractures”[Mesh] OR “Zygomatic Fractures”[Mesh] OR “Ankle Fractures”[Mesh] OR “Ankle Joint”[Mesh] OR “Ankle”[Mesh] OR fibula*[tiab] OR malleol*[tiab] OR zygo*[tiab] OR mandib*[tiab] OR orbit*[tiab] OR maxil*[tiab] OR ankl*[tiab] OR wrist*[tiab] OR radius*[tiab] OR ulna*[tiab]) AND (preform*[tiab] OR pre-form*[tiab] OR precontour*[tiab] OR contour*[tiab] OR anatomical*[tiab] OR lock*[tiab] OR 3D*[tiab] OR dimensi* [tiab]) AND (“Postoperative Complications”[Mesh] OR “Patient Satisfaction”[Mesh] OR “Length of Stay”[Mesh] OR “Operative Time”[Mesh] OR “Reoperation”[Mesh] OR reduc*[tiab] OR reposit*[tiab] OR stabil*[tiab] OR stabl*[tiab] OR complication*[tiab] OR satisfaction[tiab] OR displac*[tiab] OR reconstru* [tiab]) AND (“Controlled Clinical Trial” [Publication Type] OR “Cohort Studies”[Mesh] OR “Evaluation Study” [Publication Type] OR “Comparative Study” [Publication Type] OR controlled-study[tiab] OR controlled-trial[tiab] OR clinical-study[tiab] OR clinical-trial*[tiab] OR random*[tiab] OR prospectiv*[tiab] OR follow-up[tiab] OR cohort[tiab] OR groups[tiab] OR trial[ti] OR compar*[ti] OR evaluat*[ti] OR vs[ti] OR versus[ti]) NOT (“Review” [Publication Type] OR (“Animals”[Mesh] NOT “Humans”[Mesh]))
EMBASE	(‘fracture’/exp OR ‘fracture fixation’/exp OR fractur*:ab,ti,kw) AND (‘osteosynthesis’/exp OR ‘bone plate’/exp OR (fixat* OR osteosynth* OR plate*):ab,ti,kw) AND (‘wrist injury’/exp OR ‘wrist’/exp OR ‘radius fracture’/exp OR ‘ulna fracture’/exp OR ‘jaw fracture’/exp OR ‘orbit fracture’/exp OR ‘ankle fracture’/exp OR ‘ankle’/exp OR (fibula* OR malleol* OR zygo* OR mandib* OR orbit* OR maxil* OR ankl* OR wrist* OR radius* OR ulna*):ab,ti,kw) AND (preform* OR pre-form* OR precontour* OR contour* OR anatomical* OR lock*):ab,ti,kw AND (‘postoperative complication’/exp OR ‘complication’/de OR ‘patient satisfaction’/exp OR ‘length of stay’/exp OR ‘operation duration’/exp OR ‘reoperation’/exp OR (reduc* OR reposit* OR stabil* OR stabl* OR complication* OR satisf* OR displac*):ab,ti,kw) AND (‘controlled clinical trial’/exp OR ‘multicenter study’/exp OR ‘cohort analysis’/exp OR ‘follow up’/exp OR ‘evaluation study’/exp OR ‘comparative study’/exp OR ‘controlled study’/de OR ‘controlled-study’ OR (‘controlled-trial’ OR ‘clinical-study’ OR ‘clinical-trial*’ OR random* OR prospectiv* OR follow-up OR cohort OR groups):ab,ti OR (trial OR compar* OR evaluat* OR vs OR versus):ti) NOT (‘review’/de OR ‘conference abstract’/it OR (‘animal’/exp NOT ‘human’/exp))
COCHRANE	([mh “Fractures, Bone”] OR [mh “Fracture Fixation”] OR fractur*:ti,ab) AND ([mh “Fracture Fixation, Internal”] OR [mh “Bone Plates”] OR fixat*:ti,ab OR osteosynth*:ti,ab OR plate*:ti,ab) AND ([mh “Wrist Injuries”] OR [mh “Wrist Joint”] OR [mh Wrist] OR [mh “Radius Fractures”] OR [mh “Ulna Fractures”] OR [mh “Jaw Fractures”] OR [mh “Mandibular Injuries”] OR [mh “Orbital Fractures”] OR [mh “Zygomatic Fractures”] OR [mh “Ankle Fractures”] OR [mh “Ankle Joint”] OR [mh Ankle] OR fibula*:ti,ab OR malleol*:ti,ab OR zygo*:ti,ab OR mandib*:ti,ab OR orbit*:ti,ab OR maxil*:ti,ab OR ankl*:ti,ab OR wrist*:ti,ab OR radius*:ti,ab OR ulna*:ti,ab) AND (preform*:ti,ab OR pre-form*:ti,ab OR precontour*:ti,ab OR contour*:ti,ab OR anatomical*:ti,ab OR lock*:ti,ab)

**Table 2 jcm-12-04661-t002:** Treatment procedure and material of the plates used in the included studies. Abbreviation: N/A = not available. CT = computed tomography. CBCT = cone beam computed tomography.

Author	Year of Publication	Follow-Up (Months)	Radiographic Assessment/Imaging Technique	Conventional Plates	Material	Precontoured Plates	Material
Orbital Fractures							
Wilmowsky et al. [40]	(2020)	3 and 6 months	CT and CBCT: pre-operative and immediate, 3 and 6 months post-operatively	Non-preformed orbital floor plates	Titanium	Implant customised in size and shape based on the individual 3-dimensional template	Titanium
Sigron et al. [41]	(2020)	N/A	CT and CBCT: pre-operatively and post-operatively	Orbital floor mesh plate (MatrixMIDFACE, DePuy Synthes, Solothurn, Switzerland)	Titanium	Pre-bent plates (MatrixMIDFACE, DePuy Synthes, Solothurn, Switzerland or MODUS Midface OPS 1.5, Medartis, Basel, Switzerland)	Titanium
Zieliński et al. [42]	(2017)	1 and 6 months	CT: pre-operatively, 1 week post-operatively	Standard, intraoperatively bent mesh	Titanium	Individual implants (23 patients) or pre-bent mesh on a 3D model (16 patients)	UHMW-PE (18 patients), ZrO2 (5 patients), and titanium (16 patients)
Fan et al. [36]	(2017)	N/A	CT: pre-operatively and post-operatively	Medpor- mesh	Titanium	Medpor-Titanium mesh trimmed according to the contour of the simulated bone template	Titanium
Scolozzi [37]	(2011)	Patients were followed for at least 6 weeks	CT: pre-operatively and post-operatively	Non-preformed radial orbital mesh plate (0.3 to 0.4 mm	Titanium	3-dimensional preformed MatrixORBITAL orbital mesh (Synthes-CH 4436, Oberdorf, Switzerland)	Titanium
Momjian et al. [39]	(2011)	Patients were followed for at least 6 months	CT: pre-operatively and post-operatively	Non-pre-shaped mesh plates 0.3 mm (Synthes, CH 4436 Oberdorf Switzerland)	Titanium	Three-dimensionally preformed mesh plates 0.4 mm (Synthes, CH 4436 Oberdorf Switzerland)	Titanium
Scolozzi et al. [38]	(2010)	Patients were followed for at least 6 weeks	CT: pre-operatively and post-operatively	Non-preformed orbital mesh plates	Titanium	3D preformed orbital mesh plates	Titanium
**Upper limb fractures**							
Kong et al. [49]	(2020)	1-, 2-, 3-, and 6 months	CT-scan: pre-operatively X-ray: intraoperatively, 1-, 2-, 3-, and 6 months post-operatively	Volar plate and K-wire fixation	N/A	Pre-bending of the volar plate 3D physical fracture model	N/A
Ellwein et al. [43]	(2019)	A minimum follow-up time of 24 months was required	Radiographs: pre-operatively, intraoperatively and post-operatively	Aptus olecranon low-profile double-plate (Fa. Medartis, Basel, Switzerland)	N/A	3.5 mm olecranon locking compression plate (LCP) (Fa. DePuySynthes, Umkirchen, Germany)	N/A
Chen et al. [48]	(2019)	3, 6, and 12 months	Radiographs and CT: preoperatively Anterior-posterior and lateral X-ray: 3, 6, and 12 months postoperatively Postoperatively CT: surgeon preference	Conventional volar locking plate (FA-LCP)	N/A	2.4 mm VA-LCP low profile plating system	N/A
DelSole et al. [47]	(2016)	2 week, 6 week, 3 months and 6 months	Radiographs: pre-operatively, 2-weeks, 6-weeks, 3 months, and 6 months postoperatively	6- or 7-hole one-third tubular plate	N/A	Pre-contoured LP (Acumed (Hillsboro, OR, USA), Stryker (Kalamazoo, MI, USA), Zimmer (Warsaw, IN, USA), or Dupuy- Synthes (Paoli, PA, USA))	N/A
Shuang et al. [46]	(2016)	6 months	CT: pre-operative X-rays and 3D reconstructed CT: and post-operatively	Conventional	N/A	Osteosynthesis plates with proper sized and number of holes fabricated using a 3D printer	N/A
You et al. [45]	(2016)	A minimum follow-up time of 12 months was required	CT-scan and anteroposterior position: pre-operatively Double radiograph: 1-day post-operatively (3D group only)	Plate and screws	Steel	Preselected and prefabricated proximal humeral locking plate and screws determined by a 3D-print model simulation	Steel
Tang et al. [44]	(2012)	Immediately, 3 months and 1 year	Anteroposterior radiograph: Immediately, 3 months and 1 year postoperatively	Locking plate (Synthes, Bettlach, Switzerland)	N/A	Anatomical plate (Weigao Orthopaedic Device Co Ltd, Weihai City, China)	N/A
**Lower limb fractures**							
Zyskowski et al. [52]	(2021)	6 and 12 weeks, 6 months and 1 year	Radiographs: pre-operatively, 6 and 12 weeks, 6 months and 1 year postoperatively	DePuy Synthes^®^ one-third semitubular plate	N/A	NEWCLIP TECHNICS, Active Ankle^®^ polyaxial locking plate	N/A
Park et al. [50]	(2020)	2 and 6 weeks, 3 and 12 months	Anteroposterior and lateral radiographs: preoperatively and postoperatively	2.7 mm fixed-angle LCP (Depuy Synthes, Oberdorf, Switzerland)	N/A	2.7mm variable-angle LCP (Depuy Synthes, West Chester, PA, USA)	N/A
Bilgetekin et al. [51]	(2019)	15th day and monthly after surgery	Radiographs of the antero-posterior, lateral and mortise view: preoperatively, postoperatively and at every follow-up	Locking tubular (1/3 Tubular Locking Compression Plates ©Xrbest Jiangsu. China)	N/A	Locking anatomical plate (Distal Fibula Locking Compression Plates ©Xrbest Jiangsu. China)	N/A
Zhang et al. [56]	(2019)	Patients were followed-up for more than 12 months	CT-scan, anteroposterior X-ray or lateral X-ray: preoperatively and postoperatively	3.5-mm locking compression plate (LCP)	N/A	Plate preselected and prefabricated on the 3D-printed log-splitter injury physical model	N/A
Zheng et al. [53]	(2018)	Patients were followed-up for at least 12 months	CT: preoperative Anteroposterior and lateral X-ray: immediately after the operation, 3, 6 and 18 months postoperatively	Plates and screws	N/A	Preselected and prefabricated plate and screws determined by the 3D-printed model	N/A
Zheng et al. [54]	(2017)	Patients were followed-up for at least 12 months	CT: preoperatively Anteroposterior and oblique X-ray: immediately after the operation, 1, 3, 6, 12 and 15 months postoperatively	Plates and screws	N/A	Preselected and prefabricated plate and screws determined by the 3D-printed model	Steel
Tsukada et al. [55]	(2013)	3, 6 and 12 months	X-ray: intra-operatively, 3, 6 and 12 months postoperatively	Straight plate (LCP Metaphysical plate, Synthes Japan, Tokyo, Japan)	Titanium	Pre-shaped plate (distal fibula plate, Stryker Japan, Tokyo, Japan)	Titanium

**Table 3 jcm-12-04661-t003:** Outcome measure: stability, fractured site: orbital. Abbreviation: N: number of patients.

Parameter I: Stability and Reduction.	Anatomical 3D Placement (N/Total Population)
Orbital Fractures	
Scolozzi (2011) [37]	2D: 10/10; 100% 3D: 10/10; 100%
Scolozzi (2010) [38]	2D: 10/10; 100% 3D: 10/10; 100%

**Table 4 jcm-12-04661-t004:** Outcome measure: clinical outcome, fractured site: orbital. Abbreviation: * = statistical significance, SD: standard deviation.

Parameter II: Clinical Outcome	Operation Time (Mean min ± SD, Total Population)	Placement Time (Mean min ± SD, Total Population)	Hospitalization Period (Mean Days ± SD, Total Population)
Orbital Fractures			
Sigron (2020) [41]	2D: 99.8 ± 28.9 (12) 3D: 57.3 ± 23.4 (10) * *p* = 0.001		2D: 3.8 ± 3.0 (12) 3D: 4.6 ± 3.9 (10)
Wilmowsky (2020) [40]		2D: 11.1 ± 7.7 (11) 3D: 5.5 ± 5.4 (25) * *p* = 0.001	
Fan (2017) [36]	2D: 95.37 ± 22.19 (27) 3D: 75.34 ± 15.68 (29) * *p* < 0.05		
Zielinkski (2017) [42]	2D: Median: 100; range: 20–420 (54) 3D: Median: 80; range: 20–410 (39)		2D: Median: 4.5; range: 2–11 (54) 3D: Median: 5; range: 1–20 (39)

**Table 5 jcm-12-04661-t005:** Outcome measure: specific parameters for orbital fractures. Abbreviation: * = statistical significance, N: number of patients, SD: standard deviation.

Parameter III: Specific for the Fractured Site	Orbital Volume (Mean mL ± SD, Total Population)	Absolute Volume Difference (Mean mL ± SD, Total Population)	Maximum Fracture Collapse (Mean mm^2^ ± SD, Total Population)	Congruence of Infraorbital Rim (Complete) (N/Total Population)	Congruence of Infraorbital Rim (Good) (N/Total Population)	Congruence of Infraorbital Rim (Acceptable) (N/Total Population)	Fracture Area (Mean mm^2^ ± SD, Total Population)	Loss of Binocular Single Vision (N/Total Population)	Enopthalmos (N/Total Population)	Limitation of the Inferior Oblique Muscle (N/Total Population)
Orbital Fractures										
Sigron (2020) [41]	2D: 30.1 ± 4.2 (12) 3D: 25.7 ± 3.0 (10) *** *p* = 0.010**	2D: 1.6 ± 1.2 (12) 3D: 1.0 ± 0.7 (10)	2D: 6.9 ± 2.3 (12) 3D: 8.6 ± 5.4 (10)				2D: 408.5 ± 137.5 (12) 3D: 389.4 ± 135.1 (10)			
Wilmowsky (2020) [40]				2D: 6/11;54% 3D: 15/25;60%	2D: 6/11;54% 3D: 6/25;24%	2D: 1/11;9% 3D: 4/25;16%				
Zielinski (2017) [42]								2D: 9/54; 16% 3D: 5/39; 13%		
Scolozzi (2011) [37]									2D: 0/10;0% 3D: 0/10;0%	2D: 0/10;0% 3D: 0/10;0%
Momjian et al. (2011) [39]	2D: 21.76 (15) 3D: 20.28 (15)	2D: 0.004 (15) 3D: 0.345 (15)							2D: 0/15;0% 3D: 0/15;0%	
Scolozzi (2010) [38]	2D:19.215 (10) 3D: 21.791 (10)	2D: 0.26 (10) 3D: 0.081 (10)								
**Parameter III: Specific for the fractured site**	**Maximum width difference between fracture zone and implant** **(mean mm ± SD, total population)**	**Maximum depth difference between fracture zone and implant** **(mean mm ± SD, total population)**		**Area difference between fracture site and implant** **(mean mm^2^ ± SD, total population)**		**Angle difference in medial and inferior wall corner** **(mean ◦ ± SD, total population)**	**Enophthalmos** **(mean mm ± SD, total population)**	**Superior sulcus deformity** **(N/total population)**		**Diplopia** **(N/total population)**
**Orbital fractures**										
Fan (2017) [36]	2D: 5.60 ± 0.90 (27) 3D: 2.51 ± 0.53 (29) * *p* < 0.05	2D: 4.61 ± 0.89 (27) 3D: 2.58 ± 0.46 (29) * *p* < 0.05		2D: 84.05 ± 20.89 (27) 3D: 43.59 ± 9.53 (29) * *p* < 0.05		2D: 12.58 ± 5.04 (27) 3D: 2.82 ± 0.44 (29) * *p* < 0.05	2D: 2.5 ± 1.0 (27) 3D: 1.0 ± 0.5 (29) * *p* < 0.05	2D: 5/27; 18.5% 3D: 2/29;6.9% * *p* < 0.05		
Momjian et al. (2011) [39]										2D: 3/15; 20% 3D: 1/15; 6.6%

**Table 7 jcm-12-04661-t007:** Outcome measure: parameters specific for upper limb fractures. Abbreviation: * = statistical significance, SD: standard deviation. ^1^ Range of motion: extension/flexion and Range of motion: pronation/supination was reported in the recovery score by Chen et al. (2019). MEPS: Mayo Elbow Performance Score, MMWS: Modified Mayo Wrist score.

Parameter IV: Specific for the Fractured Site	DASH-Score (Mean ± SD, Total Population)	Pain (Mean VAS-Score ± SD, Total Population)	MEPS (mean ± SD, Total Population)	MMWS (mean ± SD, Total Population)	Range of Motion: Extension/Flexion (mean ± SD, Total Population)	Range of Motion: Pronation/Supination (mean ± SD, Total Population)
Upper Limb Fractures						
Kong (2020) [49]	2D: 24.5 ± 7.0 (16) 3D: 23.8 ± 8.1 (16)	2D: 0.9 ± 0.3 (16) 3D: 0.9 ± 0.2 (16)				
Ellwein (2019) [43]			2D: 94 ± 10 (range: 65–100) (25) 3D: 96 ± 11 (range: 60–100) (22)		2D: 127 ± 15 (80–145) (25) 3D: 130 ± 21 (range: 40–150) (25)	2D: 170 (range: 30–180) (25) 3D: 174 (range: 95–180) (25)
Chen (2019) [48]	2D: 12.8 (range: 6–18) (28) 3D: 9.2 (range: 2–12) (19) * *p* = 0.02			2D: 83.5 (range: 75–90) (28) 3D: 93.8 (range: 85–100) (19) * *p* < 0.01	2D: 82.8 % ^1^ 3D: 94.8 % ^1^ * *p* < 0.01	2D: 84.5 % ^1^ 3D: 93.8 % ^1^ * *p* < 0.01
Shuang (2016) [46]			2D: 79 ± 13 (7) 3D: 85 ± 9 (6)		2D: 93 ± 24 (7) 3D: 98 ± 27 (6)	2D: 167 ± 21 (7) 3D: 160 ± 17 (6)
Tang (2012) [44]	2D: 5.96 ± 3.48 (17) 3D: 7.04 ± 4.40 (16)					
**Parameter IV: specific for the fractured site**	**Radial height (mean mm, total population)**	**Radial inclination (mean, total population)**	**Volar tilt (mean, total population)**			
**Upper limb fractures**						
Chen (2019) [48]	2D: 20.2 (28) 3D: 19.9 (19)	2D: 10.2 (28) 3D: 10.2 (19)	2D: 8.37 (28) 3D: 7.11 (19)			

**Table 8 jcm-12-04661-t008:** Outcome measure: complications; fractured site: upper limb. Abbreviation: N: number of patients.

Parameter II: Complications	Screw Loosening (N/Total Population)	Necrosis (N/Total Population)	Infection (N/Total Population)	Nerve Injury (N/Total Population)
Upper Limb Fractures				
Kong (2020) [49]			2D: 1/16; 6.25% 3D: 0/16; 0%	2D: 0/16; 0% 3D: 0/16; 0%
Ellwein (2019) [43]	2D: 1/25; 4% 3D: 0/22; 0%			
Chen (2019) [48]		2D:0/28; 0% 3D: 0/19; 0%	2D: 0/28; 0% 3D: 0/19; 0%	
DelSole (2016) [47]			2D: 1/14; 7.14% 3D; 0/8; 0%	
Shuang (2016) [46]			2D: 0/7; 0% 3D: 0/6; 0%	

**Table 9 jcm-12-04661-t009:** Outcome measure: clinical outcome, fractured site: upper limb. Abbreviation: * = statistical significance, N: number of patients, SD: standard deviation.

Parameter III: Clinical Outcome	Revision Surgery (N/Total Population)	Hardware Failure (N/Total Population)	Cost of the Plate (Mean Dollars, Total Population)	Operation Time (Mean in min ± SD, Total Population)	Intra-Operative Blood Loss (Mean mL ± SD, Total Population)	Intraoperative Fluoroscopy (Mean Fluoroscopy Number ± SD, Total Population)	Patients Satisfaction (N/Total Population)
Upper Limb Fractures							
Kong (2020) [49]				2D: 63.5±5.9 (16) 3D: 51.4 ± 6.8 (16) * *p* < 0.001	2D: 74.2±10.3 (16) 3D: 52.3±9.9 (16) * *p* < 0.001	2D: 5.6±1.1 (16) 3D: 4.2 ± 1.3 (16) * *p* = 0.002	
Ellwein (2019) [43]	2D: 7/25; 28% 3D: 8/22; 36.36%	2D: 0/25; 0% 3D: 0/22; 0%		2D: 80 ± 29 (range: 29–150) (25) 3D: 86 ± 26 (range: 41–141) (22)			2D: 24/25; 96% 3D: 20/22; 91%
Chen (2019) [48]	2D: 2/28; 7.14% 3D: 0/19; 0%						
DelSole (2016) [47]	2D: 1/14; 7.14% 3D: 1/8; 12.5%	2D: 0/14; 0% 3D: 0/8; 0%	2D: 157.50 (14) 3D: 2071.00 (8)				
You (2016) [45]				2D: 92.03 ± 10.31 (32) 3D: 77.65 ± 8.09 (34) * *p* < 0.05	2D: 281.25 ± 57.85 (32) 3D: 235.29 ± 63.40 (34) * *p* < 0.05	2D: 10.59 ± 1.36 (32) 3D: 7.12 ± 1.57 (34) * *p* < 0.05	
Shuang (2016) [46]				2D: 92.3 ± 17.4 (7) 3D: 70.6 ± 12.1 (6) * *p* = 0.026			
Tang (2012) [44]		2D: 0/17; 0% 3D: 2/16; 0%		2D: 75.2 (range, 45–120) (17) 3D: 78.2 (range, 50–140) (16)	2D: 158.3 (range, 100–350) (17) 3D: 176.1 (range, 100–400) (17)		

**Table 10 jcm-12-04661-t010:** Outcome measure: stability; fractured site: lower limb fractures. Abbreviation: * = statistical significance, N: number of patients.

Parameter I: Stability and Reduction	Nonunion (N/Total Population)	Delayed Union (N/Total Population)	Malunion (N/Total Population)	Bone Union (N/Total Population)	Rate of Anatomic Reduction (N/Total Population)
Lower Limb Fractures					
Zyskowski (2021) [52]	2D: 0/25; 0% 3D: 0/20; 0%				
Park (2020) [50]	2D: 0/22; 0% 3D: 0/23; 0%	2D: 0/22; 0% 3D: 1/23; 4.3%	2D: 1/22; 4.5% 3D: 1/23; 4.3%		
Bilgetekin (2019) [51]	2D: 0/37; 0% 3D: 0/25; 0%				
Zhang (2019) [56]	2D: 0/13; 0% 3D: 0/16; 0%	2D: 1/13;10.5% 3D: 2/16;12.5%	2D: 1/13; 10.5% 3D: 0/16; 0%		2D: 9/13; 69.2% 3D: 13/16; 81.3%
Zheng (2018) [53]	2D: 0/48; 0% 3D: 0/45; 0%	2D: 3/48; 6.3% 3D: 2/45;4.4%	2D: 1/48; 2.1% 3D: 1/45; 2.2%		2D: 36/48; 75% 3D: 41/45; 91.1% * *p* = 0.04
Zheng (2017) [54]	2D: 0/40; 0% 3D: 0/35; 0%				
Tsukada (2013) [55]				2D: 19/21; 90.5% 3D: 23/24; 95.8%	

**Table 11 jcm-12-04661-t011:** Outcome measure: complications; fractured site: lower limb fractures. Abbreviation: N: number of patients.

Parameter II: Complications	Plate Palpable (N/Total Population)	Swelling (N/Total Population)	Deep Vein Thrombosis (N/Total Population)	Infection (N/Total Population)	Screw Loosening (N/Total Population)
Lower Limb Fractures					
Zyskowski (2021) [52]	2D: 0/25; 0% 3D: 2/20; 10%	2D: 1/25; 4% 3D: 0/20; 0%	2D: 1/25; 4% 3D: 0/20; 0%	2D: 3/25; 12% 3D: 1/20; 5%	2D: 0/25; 0% 3D: 1/20; 5%
Park (2020) [50]				2D: 0/22; 0% 3D: 1/23; 4.3%	
Bilgetekin (2019) [51]				2D: 0/37; 0% 3D: 1/25; 4%	
Zhang (2019) [56]				2D: 1/13; 10.5% 3D: 1/16; 8.3%	
Zheng (2018) [53]				2D: 4/48; 8.3% 3D: 3/45; 6.7%	
Zheng (2017) [54]				2D: 1/40; 2.5% 3D: 2/35; 5.7%	
Tsukada (2013) [55]			2D: 0/21; 0% 3D: 0/23; 0%	2D: 0/21; 0% 3D: 2/23; 8.7%	

**Table 12 jcm-12-04661-t012:** Outcome measure: clinical outcome; fractured site: lower limb fractures. Abbreviation: * = statistical significance, N: number of patients, SD: standard deviation.

Parameter III: Clinical Outcome	Removed Plates (N/Total Population)	Fracture to Union Time (Mean Months ± SD, Total Population)	Pain (Mean VAS-Score ± SD, Total Population)	Operation Time (Mean min ± SD, Total Population)	Intraoperative Fluoroscopy (Mean Fluoroscopy Number ± SD, Total Population)	Intraoperative Blood Loss (Mean mL ± SD, Total Population)
Lower Limb Fractures						
Zyskowski (2021) [52]	2D: 10/25; 40% 3D: 13/20; 65%					
Park (2020) [50]				2D: 83.5 ± 31.2 min (22) 3D: 65.8 ± 23.9 min (23) * *p* = 0.04		
Bilgetekin (2019) [51]	2D: 2/37; 5.4% 3D: 0/25; 0%					
Zhang (2019) [56]		2D: 5.2 ± 1.3 (13) 3D: 5.1 ± 1.2 (16)		2D: 124.5 ± 11.5 (13) 3D: 107.8 ± 10.2 (16) * *p* < 0.001	2D: 11.7 ± 2.4 (13) 3D: 7.3 ± 2.7 (16) * *p* < 0.001	2D: 133.7 ± 26.2 (13) 3D: 99.6 ± 19.3 (13) * *p* < 0.001
Zheng (2018) [53]		2D: 5.3 ± 1.2 (48) 3D: 5.0 ± 1.1 (45)	2D: 2.9 ± 1.2 (48) 3D: 2.6 ± 0.9 (45)	2D: 90.2 ± 10.9 (48) 3D: 74.1 ± 8.2 (45) * *p* < 0.001	2D: 11.0 ± 2.9 (48) 3D: 7.6 ± 2.2 (45) * *p* < 0.001	2D: 159.8 ± 26.5 (48) 3D: 117.1 ± 20.7 (45) * *p* < 0.001
Zheng (2017) [54]		2D: 3.2 ± 0.4 (40) 3D: 3.0 ± 0.3 (35)	2D: 2.8 ± 1.2 (40) 3D: 2.6 ± 0.9 (35)	2D: 91.3 ± 11.2 (40) 3D: 71.4 ± 6.8 (35) * *p* < 0.0001	2D: 8.6 ± 2.7 (40) 3D: 5.6 ± 1.96 (35) * *p* < 0.0001	2D: 288.7 ± 34.8 (40) 3D: 226.1 ± 22.6 (35) * *p* < 0.0001

**Table 13 jcm-12-04661-t013:** Outcome measure: parameters specific for lower limb fractures [51,52,53,54,56]. Abbreviation: * = statistical significance, N: number of patients, SD: standard deviation, AOFAS: American Orthopaedic Foot and Ankle Societ.

Parameter IV: Specific for the Fractured Site	Range of Motion: Plantarflexion (Mean ± SD, Total Population)	Range of Motion: Dorsiflexion (Mean ± SD, Total Population)	AOFAS (Mean ± SD, Total Population)	Sagittal Motion: Normal (N/Total Population)	Hindfoot Motion: Normal (N/Total Population)
Lower Limb Fractures					
Zyskowski (2021) [52]	2D: 38 ± 3 (25) 3D: 39 ± 2 (25)	2D: 22 ± 2 (25) 3D: 22 ± 3 (25)			
Zhang (2019) [56]	2D: 26.7 ± 3.4 (13) 3D: 27.9 ± 2.8 (16)	2D: 23.5 ± 3.8 (13) 3D: 24.3 ± 3.9 (16)	2D: 74.8 ± 9.3 (13) 3D: 75.5 ± 8.5 (16)		
Bilgetekin (2019) [51]			2D: Median (min-max): 87.0 (73–100) (37) 3D: Median (min-max): 85.0 (71–100) (25)	2D: 30/37; 81.1% 3D: 17/25; 68.0%	2D: 35/37; 94.6% 3D: 25/25; 100%
Zheng (2018) [53]	2D: 25.9 ± 8.7 (48) 3D: 27.4 ± 8.5 (45)	2D: 14.2 ± 5.0 (48) 3D: 15.1 ± 4.8 (45)	2D: 84.7 ± 9.0 (48) 3D: 87.4 ± 8.7 (45)		
Zheng (2017) [54]			2D: 85.8 ± 9.0 (40) 3D: 87.6 ± 7.6 (35)		

## Data Availability

Not applicable.

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
