# Peer review of "The Effectiveness of Three-Dimensional Osteosynthesis Plates versus Conventional Plates for the Treatment of Skeletal Fractures: A Systematic Review and Meta-Analysis"

_jcm, 2023, doi:10.3390/jcm12144661_

Round 1

Reviewer 1 Report

Interesting study concerning the quality/results in the treatment of fractures.

I thing that could be stressed the learning curve that is necessary in the treatment of trauma that influence not only the tretament results but also the time duration of the trìeatment 

Author Response

Thank you for your valuable comment. The reviewer is correct that there is a role for the learning curve and that the experience of the surgeon does influence the treatment outcomes and operation time.  

The subsequent sentence has been incorporated into the manuscript:

Line 484: ‘The level of experience possessed by the surgeons was not taken into consideration in this review. It needs to be noted that the experience of the surgeon can influence both the rate of anatomical reduction and the duration of the surgical procedure.’

Reviewer 2 Report

 Dear JCM, and dear authors,

Thank you very much for giving me the opportunity to review this manuscript. In general it is easy to read, and also clinically relevant, i.e., ‘new is not always better’, but in this case it is! Therefore, this review adds new information to the available literature, and in my opinion would be suitable for publication in JCM after some minor issues have been addressed:

Lines 25 + 103 + 110 + 129: the outcome measures in these locations are not completely the same. Is there a reason for this? If not, I would advise to be consistent throughout the article and change them to be exactly the same.

Line 107: It is unclear to me what exactly is meant by this protocol. Is this protocol also available?

Line 23 + 136: what about studies with exactly 10 patients? Included or excluded?

Line 33 + 39 + 159 + 375 + 377 + Table 10: what exactly is meant by the rate of anatomic reduction? What does rate mean? Only after reading the part about the Meta-analysis (line 399-416) it becomes clear to me what is meant by ‘rate’. Maybe you can clarify this a bit better in the rest of the article?

Line 200: in some boxes of Figure 1 the text is not completely readable/has fallen of. Please adjust.

Line 225: what is appendix 1? I cannot find this appendix in this article nor in the supplementary data.

Figure 3: some text in the columns (vertical rows) is not completely visible. Please adjust.

Line 287: Why is the name of this figure below the figure itself? Normally it is above the figure. Please adjust.

Line 308: it says “(Table 4)”. Is this the correct number? I think it should be “(Table 5)”.

Line 310: Is it possible to add this “p=0.002” to Table 5?

Table 5: in the first block/top left corner it says “fort he”. Please adjust to “for the”.

Line 318: in the text it says that there are no patients with limitations of the inferior oblique muscles. However, in Table 5 there are 10 out 10 with limitations, i.e., all patients had limitations. Shouldn’t it be 0 (zero) out of 10 in the Table?

Line 350 + 351: I says “in the conventional group than in the conventional group”. I think it should be “in the experimental group than in the conventional group”. Please adjust.

Line 352: is it Ellwein or Delsole? Information about the costs is information from Delsole according to/in the Table 8.

Line 488: just a remark: you could test the statistical power of your meta-analysis by a Trial Sequential Analysis. If this would be possible, I leave the decision to do this or not up to you.

Line 518: “This may have contributed to the outcomes “. Is there a possibility to check if this is true?

Discussion: it reads a bit messy. Maybe this has something to do with the number of points that are discussed, but maybe it is just me. To give an example: in lines 528-535 imaging is discussed. In one of the parts above (line 499-509) imaging is also discussed. Should it be possible to keep these parts together, i.e. one following the other? If you believe that the article reads better just as it is at this moment, I am also fine with that. Just meant a tip: a nice article to read for structuring of a manuscript/Discussion is the article of Dodson (2007): A guide for preparing a patient-oriented research manuscript. (Oral Surg Oral Med Oral Pathol Oral Radiol Endod 2007;104:307-15).

Line 552: mandibula and zygoma are not visible in Figure 1 of the article (compare with Figure 1 of the Supplementary data).

Line 557: it says “in or opinion”. I think it should be “in our opinion”. Please adjust.

Good luck with adjusting the manuscript, and keep up the good work!

Author Response

Thank you very much for giving me the opportunity to review this manuscript. In general it is easy to read, and also clinically relevant, i.e., ‘new is not always better’, but in this case it is! Therefore, this review adds new information to the available literature, and in my opinion would be suitable for publication in JCM after some minor issues have been addressed:

Thanks for your constructive remarks and valuable feedback. We have endeavored to improve and enhance the manuscript.

Lines 25 + 103 + 110 + 129: the outcome measures in these locations are not completely the same. Is there a reason for this? If not, I would advise to be consistent throughout the article and change them to be exactly the same.

Thank you for your valuable remark. Outcome measures included anatomical reduction, stability, operation time, hospitalization days, patients’ outcomes and complications.

For that reason we made a change in the following sentences:

Line 103: ‘Therefore, the present systematic review aimed to evaluate whether preformed anatomically shaped osteosynthesis plates and PSIs are preferable over conventional flat plates to treat skeletal fractures in terms of anatomical reduction, stability, operation time, hospitalisation days, patients outcomes and complications.’

Line 110: ‘Are there outcome differences between preformed anatomically shaped osteosynthesis plates (both stock implants and PSIs) and conventional flat plates in the management of fractures, especially regarding anatomical reduction, stability, operation time, hospitalisation days, patients outcomes and complications.’

Line 129: ‘Secondary Outcomes: stability and complications, days spent in hospital, operation time, patient satisfaction’

Line 107: It is unclear to me what exactly is meant by this protocol. Is this protocol also available?

When a systematic review is written, it is important that the authors make a protocol a priori. This protocol outlines the methodology: the inclusion criteria, screening process, risk of bias assessment and handling potential ambiguous cases. The methodology section of our systematic reviews provides a detailed description of our protocol.

Line 23 + 136: what about studies with exactly 10 patients? Included or excluded?

Studies with exactly 10 patients were also included. We tried to explain this better by chancing the following lines:

Line 22: ‘Eligible studies were randomized clinical trials, prospective controlled clinical trials, and prospective and retrospective cohort studies (n ≥ 10).’

Line 136: ‘studies with fewer than 10 patients.’

Line 33 + 39 + 159 + 375 + 377 + Table 10: what exactly is meant by the rate of anatomic reduction? What does rate mean? Only after reading the part about the Meta-analysis (line 399-416) it becomes clear to me what is meant by ‘rate’. Maybe you can clarify this a bit better in the rest of the article?

We agree with you that the definition of rate of anatomic reduction and stability can be clarified.

We inserted the following Line (162) in the manuscript: ‘In this review the rate of anatomic reduction refers to a percentage of fractures that have been accurately repositioned to their anatomically correct alignment. Stability is defined as ‘a reduced ZMC fracture, which remains in a stable anatomical position over time’.’

Line 200: in some boxes of Figure 1 the text is not completely readable/has fallen of. Please adjust.

Thanks a lot for your remark, we adjusted Figure 1.

Line 225: what is appendix 1? I cannot find this appendix in this article nor in the supplementary data.

Thanks for your remark. We changed ‘Appendix 1’ to ‘Table 2’ in Line 225.

Figure 3: some text in the columns (vertical rows) is not completely visible. Please adjust.

We adjusted Figure 3. The high quality version of Figure 3 can be found in the supplementary file. 

Line 287: Why is the name of this figure below the figure itself? Normally it is above the figure. Please adjust.

We corrected this by inserting the name above the figure (Line 280).

Line 308: it says “(Table 4)”. Is this the correct number? I think it should be “(Table 5)”.

We changed ‘Table 4’ to ‘Table 5’ in Line 308.

Line 310: Is it possible to add this “p=0.002” to Table 5?

Table 5 displays the differences between the conventional and experimental group. If a significance level is reported in this table, this means that there is a significance difference found between the two groups (conventional vs experimental). Sigron et al. (2020) reported a statistically significant difference (p = 0.002) in the mean absolute volume of the preoperative non-fractured orbital and the postoperative reconstructed orbital in the conventional group. This means that the results only depict on the conventional group. In our opinion it would be confusing to add this to Table 5, since no comparison was made between the two groups.

Table 5: in the first block/top left corner it says “fort he”. Please adjust to “for the”.

We changed the first block of Table 5: ‘fort he’ into ‘for the’.

Line 318: in the text it says that there are no patients with limitations of the inferior oblique muscles. However, in Table 5 there are 10 out 10 with limitations, i.e., all patients had limitations. Shouldn’t it be 0 (zero) out of 10 in the Table?

We changed limitations of the inferior oblique muscles in 0/10;0% in Table 5.

Line 350 + 351: I says “in the conventional group than in the conventional group”. I think it should be “in the experimental group than in the conventional group”. Please adjust.

We changed the following line 350 +351: ‘Both operation time and intraoperative blood loss and intraoperative fluoroscopy times were significantly shorter and lower in the experimental group than in the conventional group.’

Line 352: is it Ellwein or Delsole? Information about the costs is information from Delsole according to/in the Table 8.

We removed line 352 from our manuscript.

Line 488: just a remark: you could test the statistical power of your meta-analysis by a Trial Sequential Analysis. If this would be possible, I leave the decision to do this or not up to you.

Thanks a lot for you suggestion. We consulted with our statisticus regarding the appropriate statistical tests for our study. It was determined that the Trial Sequential Analysis is not indicated for our study.

Line 518: “This may have contributed to the outcomes “. Is there a possibility to check if this is true?

Thank you very much for your input. We have indeed considered this aspect and acknowledge it as a limitation, which is thoroughly explained in our discussion. For this study, we investigated whether a preformed shape contributes to better treatment outcomes. By preformed shape, we refer to a plate that has a predetermined shape pre-operative (the experimental 3D group), which includes both patient-specific implants (PSIs) and preformed plates based on the statistical plate model. In this study, it is not possible to differentiate within the 3D group, because studies within this group use PSI’s and preformed anatomical plates.

This study provides initial insights into the differences between conventional plates and 3D plates with a specific preformed shape. Conducting a future study that compares: 1) conventional flat plates, 2) preformed plates, and 3) PSI’s would be of great value and provide further insights into their comparative effectiveness.

Discussion: it reads a bit messy. Maybe this has something to do with the number of points that are discussed, but maybe it is just me. To give an example: in lines 528-535 imaging is discussed. In one of the parts above (line 499-509) imaging is also discussed. Should it be possible to keep these parts together, i.e. one following the other? If you believe that the article reads better just as it is at this moment, I am also fine with that. Just meant a tip: a nice article to read for structuring of a manuscript/Discussion is the article of Dodson (2007): A guide for preparing a patient-oriented research manuscript. (Oral Surg Oral Med Oral Pathol Oral Radiol Endod 2007;104:307-15).

Thank you for your tip, it is indeed a nice article to read for structuring of a manuscript.

We understand that imaging is twice discussed in the discussion.

The first passage (499-509) primarily focuses on the benefits of digital planning using three-dimensional pre-operative imaging. These advantages include providing the surgeon with substantial information regarding the fracture pattern pre-operative and per-operative, thereby facilitating a more efficient and improved surgical procedure.

The second passage (528-535) mainly revolves around the distinction in algorithms between the included studies. This disparity in algorithms, for instance, the absence of post-operative imaging in one study while being present in another. This variation in algorithms makes it challenging to truly compare the treatment outcomes between them.

Hence, these two aspects (passage 1: 499-509 and passage 2: 528-535) form separate independent discussion points.

Line 552: mandibula and zygoma are not visible in Figure 1 of the article (compare with Figure 1 of the Supplementary data).

We corrected this by inserting mandibular and zygoma in Figure 1 of the article.

Line 557: it says “in or opinion”. I think it should be “in our opinion”. Please adjust.

Thanks for your constructive remark. We changed the ‘in or opinion’ into ‘in our opinion’ in line 557.

Good luck with adjusting the manuscript, and keep up the good work! 

Thanks a lot for all your good suggestions!

Reviewer 3 Report

The present study is interesting and methodologically well structured. IPS preformed plates play an increasingly important role in maxillofacial surgery, and it is important to know their applications and advantages. However, to analyze in the same study orbital fractures with lower and upper limb fractures, I consider that it is not appropriate because it has a very different behavior and treatment strategy.

Author Response

Dear reviewer, thank you for your valuable comment. We agree with the reviewer's observation that there exists distinct behavioral patterns and treatment strategies for each extremity and fracture. However, this review focuses specifically on the comparison of anatomic repositioning and stability between preformed anatomical plates and conventional plates in traumatology. We have deliberately expanded our investigation beyond maxillofacial fractures to encompass the potential benefits of preformed anatomical plates in the treatment of skeletal fractures comprehensively.

It is often suggested that 3D plates offer superior outcomes for the treatment of skeletal fractures. The incentive for this review was to investigate, whether this is scientifically substantiated. This study provides an overview of the initial assessment of the existing literature. In future studies, it would undoubtedly be valuable to conduct specialized assessments and analyses for each fracture type within specific fields of expertise.

Round 2

Reviewer 3 Report

Dear authors,

This work is part of a Special Issue on Maxillofacial Surgery and I find no sense in reviewing fractures of the orbit together with fractures of the upper and lower limb. The biological and functional behavior of these fractures together with their repercussions and complications have nothing to do with each other.

For this reason, I consider that a major revision should be carried out taking into account only fractures of the orbital region without comparing them with fractures of the upper and lower limb.

Author Response

26-05-2023

Amsterdam

Dear editor,

We are writing in response to the reviewer's comment and suggestion put forth for our manuscript: “Treatment of skeletal fractures: anatomically preformed osteosynthesis plates and patient-specific implants versus conventional plates - effect on reposition and clinical outcomes. A systematic review and meta-analysis”. First and foremost, we would like to thank the reviewer for the time invested in reviewing this manuscript and the valuable suggestion that is provided. A comprehensive response to the comment is found below.  

Reviewer 1  

Comment to authors:  

Dear authors,

This work is part of a Special Issue on Maxillofacial Surgery and I find no sense in reviewing fractures of the orbit together with fractures of the upper and lower limb. The biological and functional behavior of these fractures together with their repercussions and complications have nothing to do with each other.

For this reason, I consider that a major revision should be carried out taking into account only fractures of the orbital region without comparing them with fractures of the upper and lower limb.

Dear reviewer,

We sincerely appreciate your valuable comment on our manuscript. We fully acknowledge and agree with the reviewer's observation regarding the existence of distinct behavioral patterns and treatment strategies for each extremity and fracture. However, it is crucial to emphasize that the scope of this review is specifically centered around comparing the anatomic repositioning and clinical outcomes achieved by utilizing three-dimensional preformed anatomical plates as opposed to conventional plates in the field of traumatology.

The primary focus lies on the comparison between three-dimensional plates and conventional flat plates in the treatment of skeletal fractures, without delving into the intricacies of fracture behavior or characteristics. The outcomes of this systematic review highlight general outcome variables, such as operation time and infection, while also encompassing specific complications associated with each fracture type, thereby ensuring a comprehensive assessment. It is important to note that the analysis was strictly kept separate, with each of the three groups (orbita, upper limb, and lower limb) being analyzed independently. The analysis of orbita fractures was conducted independently from the analysis of upper and lower limb fractures, and so forth. The groups were not compared to each other, and the results were kept separate. This choice was made specifically to ensure the preservation of the differentiation among various types of fractures, thus maintaining the integrity of their unique characteristics.

The primary motivation for undertaking this systematic review was to investigate the scientific substantiation of the commonly suggested notion that three-dimensional plates offer superior treatment outcomes for skeletal fractures. We purposefully expanded the scope of our investigation beyond maxillofacial fractures to encompass a comprehensive assessment of the potential benefits offered by preformed anatomical plates in the treatment of skeletal fractures.

This study presents an overview of the initial assessment of the existing literature, shedding light on the current state of knowledge in the traumatology. Considering only orbital fractures in this review would significantly alter our main research question. It is undoubtedly crucial for future prospective or retrospective studies to conduct specialized assessments and analyses tailored to each fracture type within specific domains of expertise. Such endeavors would undoubtedly provide valuable insights.

We have modified the title of the manuscript to provide a more accurate representation of the article: ‘Effectiveness of three-dimensional osteosynthesis plates versus conventional plates for the treatment of skeletal fractures: A systematic review and meta-analysis.’.

Once again, we express our gratitude for your contribution and valuable feedback. We sincerely hope that the manuscript, along with this improvement, is now suitable for publication in JCM.   

Yours faithfully, 

I.I. Raghoebar and prof. L. Dubois
